



# Error estimate for fluxgate magnetometer in-flight calibration on a spinning spacecraft

Yasuhito Narita[1], Ferdinand Plaschke[1], Werner Magnes[1], David Fischer[1], and Daniel Schmid[1]

[1]Space Research Institute, Austrian Academy of Sciences, Schmiedlstr. 6, A-8042 Graz, Austria

**Correspondence:** Y. Narita
(yasuhito.narita@oeaw.ac.at)

**Abstract.** Fluxgate magnetometers are widely used for in-situ magnetic field measurements in the context of geophysical and solar system studies. Like in most of experimental studies, magnetic field measurements using the fluxgate magnetometers are constrained to the associated uncertainties. To evaluate the performance of magnetometers, the measurement uncertainties of calibrated magnetic field data are quantitatively studied for a spinning spacecraft. The uncertainties are derived analytically by perturbing the calibration procedure, and are simplified into the first-order expression including the offset errors and the coupling of calibration parameter errors with the ambient magnetic field. The error study shows how the uncertainty sources combine through the calibration process. The final error depends on the ambient environment such as the magnitude of magnetic field relative to the offset error and the angle of magnetic field to the spacecraft spin axis are important factors. The offset uncertainties are the major factor in a low-field environment, while the angle uncertainties (rotation angle in the spin plane, sensor non-orthogonality, and sensor misalignment to the spacecraft reference directions) become more important in a high-field environment in a proportional way to the magnetic field. The error formulas serve as a useful tool in designing high-precision magnetometers in future spacecraft missions as well as in data analysis methods in geophysical and solar system science.

## 1 Introduction

Fluxgate magnetometers perform measurements from DC (direct current) to low-frequency magnetic field vectors (typically up to 10–100 Hz), and are widely applied to in situ spacecraft observations for space plasma, magnetospheric, and heliospheric research (Acuña, 2002). The fluxgate magnetometers can be mounted on a spinning spacecraft or three-axis stabilized one, depending on the individual mission concept. In particular, in-flight calibration benefits from the spacecraft spin, since 8 of 12 calibration parameters are determined by making use of the spacecraft spin. Detailed procedure for the in-flight calibration on a spinning spacecraft are presented by, e.g., Kepko et al. (1996) and Plaschke et al. (2019).

The goal of the current paper is to give an outline of systematic errors of calibrated fluxgate magnetometer data on a spinning spacecraft. The error of magnetic field data occurs due to the uncertainties of the calibration parameters. The error sources may combine with one another through the calibration process. We derive the full expression of calibration errors as well as a more practical, simplified expression by truncating at the first order of relative errors.





## 2 Systematic error on in-flight calibration

For a spin-stabilized spacecraft, the magnetometer in-flight calibration is performed by correcting for offsets (including the spacecraft DC field), gains, deviations from the ideal orthogonal coordinate system, spacecraft spin axis direction with respect to the sensor reference direction and rotation angle around the spacecraft spin axis. For a nearly-orthogonal unit-gain sensor system, the measured magnetic field is transformed into a de-spun coordinate system, and is expanded into a Fourier series of spacecraft spin frequency and its harmonics.

The magnetic field vector measured by the three sensors (sensor ourput) is related to the ambient field by taking account of spacecraft spin-axis direction, spacecraft spin phase, sensor-axis directions, sensitivities (or gains) of the sensors, and offsets (Kepko et al., 1996; Plaschke et al., 2019). The relation is constructed in the following fashion.

1. The true or model ambient field is set in the inertial (i.e., non-spinning) orthogonal spacecraft spin axis-aligned coordinate system (the coord-1 system in Fig. 1) with the spin-plane component in the X direction ($B_X = B_p$) and the spin-axis component in the Z direction ($B_Z = B_a$). There is no magnetic field in the rest spin-plane component, $B_Y = 0$, because the coord-1 system spans the spacecraft spin axis (in the Z direction) and the ambient field in the X–Z plane.

2. The model ambient field in the coord-1 system is transformed into the spinning orthogonal spin-axis-aligned system (the coord-2 system in Fig. 1) with the magnetic field components $B_x$, $B_y$, and $B_z$ by refering to the spin axis as the z direction and rotating the spin plane around the spin axis by the spacecraft spin phase $-\omega t$ (here $\omega$ is defined as the de-spinning frequency and $-\omega$ as the spin frequency; $t$ the time) as

$$B_x = B_X \cos(-\omega t) \tag{1}$$
$$B_y = B_Y \sin(-\omega t) \tag{2}$$
$$B_z = B_Z. \tag{3}$$

3. The field is then transformed into the spinning, orthogonal sensor package system (the coord-3 system in Fig. 1) by further rotating around the spin axis by correcting for the magnetometer boom extension and a possible misalignment of the fluxgate sensor in the spin plane (with the rotation angle $\phi_a$) and orienting the Pz axis in the sensor-3 direction with the spin axis tilt angles $\sigma_{Px}$ and $\sigma_{Py}$ (with respect to the Pz axis) to obtain the magnetic field components as $B_{Px}$, $B_{Py}$, and $B_{Pz}$ (here, P in the subscript stands for the sensor package).

4. The field is further transformed into the spinning, non-orthogonal sensor-axis-aligned system (the coord-4 system in Fig. 1) by correcting for the elevation angles $\theta_1$ (between the sensor-1 and the sensor-3 directions) and $\theta_2$ (between the sensor-2 and the sensor-3 directions) and also for the azimuthal separation angle $\phi_{12}$ (between the sensor-1 and sensor-2 projected onto the plane normal to the sensor-3 direction) to obtain the magnetic field components $B_1$, $B_2$, and $B_3$ in the directions of the sensor axes including the gains and the offsets.





5. Finally, in the calibration procedure, the above transformations are inverted to estimate the ambient field from the sensor output. The estimated or reconstructed field is expressed the de-spun inertial coordinate system (the coord-5 system in Fig. 1) with the spin-plane primary component ($B_{X'}$), spin-plane residual component ($B_{Y'}$), and spin-axis component ($B_{Z'}$). The primed field expression in the coord-5 system ($B_{X'}$, $B_{Y'}$, and $B_{Z'}$) is identical to the model ambient field $B_X$, $B_Y$, and $B_Z$ in the coord-1 system if the calibration parameters are all accurately known.

Note that the forward transformation is defined for the conversion of the sensor output (in the coord-4 system) into the magnetic field in the physically relevant system (the coord-1 system). In the error estimate study, the inverse transformation from the coord-1 system to the coord-4 system is more instructive in order to compare the calibrated magnetic field vector in the coord-5 system with the model ambient field in the coord-1 system.

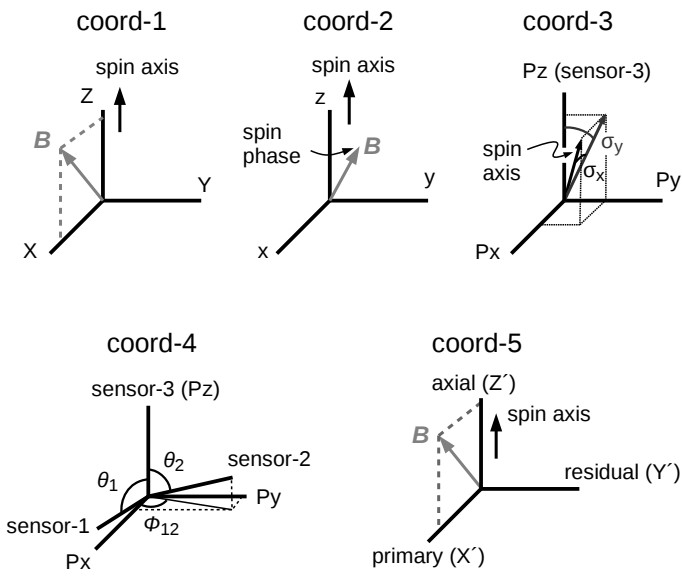

**Figure 1.** Coordinate systems used in the magnetometer calibration error estimate.

The relation between the sensor-output magnetic field $\boldsymbol{B}_s$ (introduced in the coord-4 system) and the model ambient

field in the spinning frame $\boldsymbol{B}$ (introduced in the coord-2 system, Eqs. 1–3) is expressed by a set of transformation matrices $\mathbf{G}^{-1}\,\boldsymbol{\Gamma}^{-1}\,\boldsymbol{\Sigma}^{-1}\,\boldsymbol{\Phi}^{-1}$ and an offset vector $\boldsymbol{O}_s$ as (Plaschke et al., 2019)

$$\boldsymbol{B}_s = \mathbf{G}^{-1}\,\boldsymbol{\Gamma}^{-1}\,\boldsymbol{\Sigma}^{-1}\,\boldsymbol{\Phi}^{-1}\boldsymbol{B} + \boldsymbol{O}_s. \tag{4}$$

Here, the set of transformation matrices is composed of (1) the inverse rotation matrix around the spin axis $\boldsymbol{\Phi}^{-1}$ by the rotation angle $\phi_a$, (2) the inverse rotation matrix $\boldsymbol{\Sigma}^{-1}$ correcting for the tilt of spacecraft spin axis to the Pz direction (transforming

the coord-2 system into the coord-3 system), (3) the inverse conversion matrix $\boldsymbol{\Gamma}^{-1}$ (transforming the coord-3 system into the coord-4 system) and (4) the inverse gain matrix $\mathbf{G}^{-1}$. The sensor-output field is then corrected for the offset vector $\boldsymbol{O}_s$ in the





sensor-axis directions. There matrices are constructed as follows (Plaschke et al., 2019).

$$\mathbf{\Phi}^{-1} = \begin{pmatrix} 1 & \phi_{\mathrm{a}} & 0 \\ -\phi_{\mathrm{a}} & 1 & 0 \\ 0 & 0 & 1 \end{pmatrix} \tag{5}$$

$$\mathbf{\Sigma}^{-1} = \begin{pmatrix} 1 & 0 & \sigma_{\mathrm{Px}} \\ 0 & 1 & \sigma_{\mathrm{Py}} \\ -\sigma_{\mathrm{Px}} & -\sigma_{\mathrm{Py}} & 1 \end{pmatrix} \tag{6}$$

$$\mathbf{\Gamma}^{-1} = \begin{pmatrix} 1 & 0 & -\delta\theta_1 \\ -\delta\phi_{12} & 1 & -\delta\theta_2 \\ 0 & 0 & 1 \end{pmatrix} \tag{7}$$

$$\mathbf{G}^{-1} = \begin{pmatrix} (gG_{\mathrm{p}})^{-1} & 0 & 0 \\ 0 & g^{-1}G_{\mathrm{p}} & 0 \\ 0 & 0 & G_{\mathrm{a}}^{-1} \end{pmatrix}. \tag{8}$$

The calibrated magnetic field vectors depend on the ambient magnetic field ($B_{\mathrm{p}}$ in the spin plane and $B_{\mathrm{a}}$ along the spin axis) and the following calibration parameters:

- gain ratio $g$ between the two spin-plane sensors

- absolute gains in the spin plane $G_{\mathrm{p}}$ and that in the spin axis direction $G_{\mathrm{a}}$

- offsets in the three sensor directions, $O_1$, $O_2$, and $O_3$

- spin axis tilt angles $\sigma_{\mathrm{Px}}$ and $\sigma_{\mathrm{Py}}$ to the angles in sensor package system (angle between the coord-2 system and the coord-3 system)

- elevation angles $\theta_1$ and $\theta_2$ with a relation to the deviation from 90 degree, $\delta\theta_1 = \theta_1 - 90°$ and $\delta\theta_2 = \theta_2 - 90°$ for the sensors 1 and 2, respectively

- azimuthal angle $\phi_{12}$ with a relation to the deviation from 90 degree, $\delta\phi_{12} = \phi_{12} - 90°$

- rotation angle $\phi_{\mathrm{a}}$ in the spin plane

Note that the orthogonality nearly holds such that the elevation and azimuthal angles exhibit only a small deviation from 90 degree,

$$\delta\theta_1 = \theta_1 - \frac{\pi}{2} \sim 0 \tag{9}$$

$$\delta\theta_2 = \theta_2 - \frac{\pi}{2} \sim 0 \tag{10}$$

$$\delta\phi_{12} = \phi_{12} - \frac{\pi}{2} \sim 0. \tag{11}$$



Also the tilt angles are small and close to zero,

$$\sigma_{Px} \quad \sim \quad 0 \tag{12}$$

$$\sigma_{Py} \quad \sim \quad 0. \tag{13}$$

The relative gain and the two absolute gains are close to unity,

$$g \quad \sim \quad 1 \tag{14}$$

$$G_p \quad \sim \quad 1 \tag{15}$$

$$G_a \quad \sim \quad 1. \tag{16}$$

The sensor output in the de-spun coordinate system (including the temperature dependence) is expressed up to the second lowest-order of the spin frequency as (Eqs. 24–26 in Plaschke et al., 2019):

$$
\begin{aligned}
B_{X'} \quad = \quad & \frac{B_p(1+g^2)}{2gG_p} \\
& + \left[ O_1 + \frac{B_a(\sigma_{Px} - \delta\theta_1)}{gG_p} \right] \cos\omega t \\
& - \left[ O_2 + \frac{gB_a(\sigma_{Py} - \delta\theta_2)}{G_p} \right] \sin\omega t \\
& + \left[ \frac{B_p(1-g^2)}{2gG_p} \right] \cos 2\omega t \\
& + \frac{B_p}{2G_p} \left[ g\phi_a - \frac{\phi_a}{g} + g\delta\phi_{12} \right] \sin 2\omega t
\end{aligned}
\tag{17}
$$

$$
\begin{aligned}
B_{Y'} \quad = \quad & -\frac{B_p}{2G_p} \left[ \frac{1+g^2}{g}\phi_a + g\delta\phi_{12} \right] \\
& + \left[ O_2 + \frac{gB_a(\sigma_{Py} - \delta\theta_2)}{G_p} \right] \cos\omega t \\
& + \left[ O_1 + \frac{B_a(\sigma_{Px} - \delta\theta_1)}{gG_p} \right] \sin\omega t \\
& - \frac{B_p}{2G_p} \left[ g\phi_a - \frac{\phi_a}{g} + g\delta\phi_{12} \right] \cos 2\omega t \\
& + \left[ \frac{B_p(1-g^2)}{2gG_p} \right] \sin 2\omega t
\end{aligned}
\tag{18}
$$

$$
B_{Z'} \quad = \quad \frac{B_a}{G_a} + O_3 - \frac{B_p\sigma_{Px}}{G_a}\cos\omega t + \frac{B_p\sigma_{Py}}{G_a}\sin\omega t \tag{19}
$$

Here, the magnetic field vector $(B_{X'}, B_{Y'}, B_{Z'})$ is represented in the coord-5 system in which $B_{Z'}$ is the component of ambient field along the spacecraft spin axis (hereafter the spin-axis component), $B_{X'}$ is the spin-plane component of ambient field reproduced by the calibration procedure (the spin-plane primary component), and $B_{Y'}$ is in the spin-plane but does not contain the reproduced ambient field direction (the spin-plane residual component). If the calibration parameters were all accurately





known, the residual component ($B_{Y'}$) would be zero, and the ambient field reproduced or reconstructed by the calibration has
the spin-plane component ($B_{X'}$) and the spin-axis component ($B_{Z'}$). The directions of the three components are orthogonal if
the calibration is accurate. Non-orthogonality may arise due to the uncertainties in the calibration parameters. The spacecraft
spin frequency $\omega$ (as angular frequency) is assumed to be well known. $t$ denotes the time in Eqs. (17)–(19). We also assume
that the calibration parameters do not change over the time or along the spacecraft orbit.

## 2.1 Spin-plane primary component

The systematic error of magnetic field data is analytically derived by perturbing the calibration equations (Eqs. 17–19). The
error in the X$'$ component (spin-plane primary component) is denoted by $\Delta B_{X'}$. The spin-plane primary component is assumed
to be aligned with the ambient field direction in the spin plane after calibration. On the assumption of the constant spin
frequency ($\omega = \mathrm{const.}$), the error $\Delta B_{X'}$ is derived by perturbing Eq. (17) as follows:

$$
\begin{aligned}
|\Delta B_{X'}| \quad \le \quad & \max\left(\Delta O_1,\, \Delta O_2\right) + B_{\mathrm{p}}\, \Delta\left(\frac{1}{2G_{\mathrm{p}}}\left(\frac{1}{g}+g\right)\right) \\
& + B_{\mathrm{a}} \max\left(\Delta\left(\frac{1}{gG_{\mathrm{p}}}\,|\sigma_{\mathrm{Px}}-\delta\theta_1|\right),\, \Delta\left(\frac{g}{G_{\mathrm{p}}}\,|\sigma_{\mathrm{Py}}-\delta\theta_2|\right)\right) \\
& + B_{\mathrm{p}} \max\left(\Delta\left(\frac{1}{2G_{\mathrm{p}}}\left|\frac{1}{g}-g\right|\right),\, \Delta\left(\frac{1}{2G_{\mathrm{p}}}\left|\frac{1}{g}-g\right|\phi_{\mathrm{a}}+g\,\delta\phi_{12}\right)\right)
\end{aligned}
\tag{20}
$$

Here, the function $\max(x, y)$ returns the larger value from two variables, $x$ and $y$, and is defined as

$$
\max(x, y) = \frac{1}{2}\left(x + y + |x - y|\right)
\tag{21}
$$

The function $\max(x, y)$ takes the largest amplitude from an elliptically-shaped time series signal such as $x\cos(\omega t) + y\sin(\omega t)$.
After differential calculus (see Appendix), the expression of error $\Delta B'_{X}$ is arranged to that of calibration parameters (gains,





offsets, and angles):

$$
\begin{aligned}
|\Delta B_{X'}| \leq \ & \max\left(\Delta O_1,\, \Delta O_2\right) \\
& + B_\mathrm{p}\, \frac{1}{2G_\mathrm{p}^2}\left[\left(\frac{1}{g}+g\right)+\left|\frac{1}{g}-g\right|\max\left(1,\,\phi_\mathrm{a}\right)\right]\Delta G_\mathrm{p} \\
& + B_\mathrm{a}\, \frac{1}{G_\mathrm{p}^2}\max\left(\frac{1}{g}\left|\sigma_\mathrm{Px}-\delta\theta_1\right|,\, g\left|\sigma_\mathrm{Py}-\delta\theta_2\right|\right)\Delta G_\mathrm{p} \\
& + B_\mathrm{p}\left[\frac{1}{2G_\mathrm{p}}\left|1-\frac{1}{g^2}\right|+\frac{1}{2G_\mathrm{p}}\left(\frac{1}{g^2}+1\right)\max\left(1,\,\phi_\mathrm{a}\right)+\delta\phi_{12}\right]\Delta g \\
& + B_\mathrm{a}\, \frac{1}{G_\mathrm{p}}\max\left(\frac{1}{g^2}\left|\sigma_\mathrm{Px}-\delta\theta_1\right|,\, \left|\sigma_\mathrm{Py}-\delta\theta_2\right|\right)\Delta g \\
& + B_\mathrm{a}\, \frac{1}{G_\mathrm{p}}\max\left(\frac{\Delta\sigma_\mathrm{Px}}{g},\, g\,\Delta\sigma_\mathrm{Py}\right) \\
& + B_\mathrm{a}\, \frac{1}{G_\mathrm{p}}\max\left(\frac{\Delta(\delta\theta_1)}{g},\, g\,\Delta(\delta\theta_2)\right) \\
& + B_\mathrm{p}\, \frac{1}{2G_\mathrm{p}}\left|g-\frac{1}{g}\right|\Delta\phi_\mathrm{a} \\
& + B_\mathrm{p}\, g\,\Delta(\delta\phi_{12}).
\end{aligned}
\tag{22}
$$

It is useful to introduce the following variables to simplify the notations:

$$
\begin{aligned}
\Delta O_{\mathrm{S1/2}} &= \max\left(\Delta O_1, \Delta O_2\right) && (23) \\
\Delta\sigma_{\mathrm{Px/y}} &= \max\left(\Delta\sigma_\mathrm{Px}, \Delta\sigma_\mathrm{Py}\right) && (24) \\
\Delta(\theta_{\mathrm{S1/2}}) &= \max\left(\Delta(\delta\theta_1), \Delta(\delta\theta_2)\right) && (25)
\end{aligned}
$$

If the gains (both absolute and relative ones) are close to unity ($g \simeq 1$, $G_\mathrm{p} \simeq 1$) and the misalignments are small ($\sigma_\mathrm{Px} \ll 1$ rad, $\sigma_\mathrm{Py} \ll 1$ rad, $\delta\theta_1 \ll 1$ rad, $\delta\theta_2 \ll 1$ rad, $\delta\phi_{12} \ll 1$ rad), Eq. (22) is simplified with the leading terms:

$$
\begin{aligned}
|\Delta B_{X'}| \leq \ & \Delta O_{\mathrm{S1/2}} \\
& + B_\mathrm{p}\left(\Delta G_\mathrm{p} + \max(1,\,\phi_\mathrm{a})\Delta g + \Delta(\delta\phi_{12})\right) \\
& + B_\mathrm{a}\left(\Delta\sigma_{\mathrm{Px/y}} + \Delta(\delta\theta_{\mathrm{S1/2}})\right).
\end{aligned}
\tag{26}
$$

We assume $\max(1,\,\phi_\mathrm{a}) = 1$ (which is realized when $\phi_\mathrm{a} \leq 1$ holds), then Eq. (26) is further simplified into a more practical form:

$$
\begin{aligned}
|\Delta B_{X'}| \leq \ & \Delta O_{\mathrm{S1/2}} \\
& + B_\mathrm{p}\left(\Delta G_\mathrm{p} + \Delta g + \Delta(\delta\phi_{12})\right) \\
& + B_\mathrm{a}\left(\Delta\sigma_{\mathrm{Px/y}} + \Delta(\delta\theta_{\mathrm{S1/2}})\right).
\end{aligned}
\tag{27}
$$



## 2.2 Spin-plane residual component

Derivation of the error in the $Y'$ component (which residual to the primary component after determination or reconstruction of the ambient field in the spin plane) nearly follows that in the $X'$ component. Note that the $Y'$ component has only a tiny amount of the ambient field because of its residual character. The $Y'$ component vanishes if the calibration is properly and accurately done. After derivative calculations (see Appendix), the error of the residual component is estimated as

$$
\begin{aligned}
|\Delta B_{Y'}| \leq\ & \max\left(\Delta O_1,\, \Delta O_2\right) \\
& + B_{\mathrm{p}}\, \Delta\left(\frac{1}{2G_{\mathrm{p}}}\left(\left(\frac{1}{g}+g\right)\phi_{\mathrm{a}}+g\,\delta\phi_{12}\right)\right) \\
& + B_{\mathrm{a}}\max\left(\Delta\left(\frac{1}{gG_{\mathrm{p}}}|\sigma_{\mathrm{Px}}-\delta\theta_1|\right),\, \Delta\left(\frac{g}{G_{\mathrm{p}}}|\sigma_{\mathrm{Py}}-\delta\theta_2|\right)\right) \\
& + B_{\mathrm{p}}\max\left(\Delta\left(\frac{1}{2G_{\mathrm{p}}}\left|\frac{1}{g}-g\right|\right),\, \Delta\left(\frac{1}{2G_{\mathrm{p}}}\left|\frac{1}{g}-g\right|\phi_{\mathrm{a}}+g\,\delta\phi_{12}\right)\right)
\end{aligned}
\tag{28}
$$

Equation (28) is sorted to the errors of calibration parameters as:

$$
\begin{aligned}
|\Delta B_{Y'}| \leq\ & \max\left(\Delta O_1,\, \Delta O_2\right) \\
& + B_{\mathrm{p}}\,\frac{1}{2G_{\mathrm{p}}^2}\left[\left(\frac{1}{g}+g\right)\phi_{\mathrm{a}}+\left|\frac{1}{g}-g\right|\max\left(1,\,\phi_{\mathrm{a}}\right)\right]\Delta G_{\mathrm{p}} \\
& + B_{\mathrm{a}}\,\frac{1}{G_{\mathrm{p}}^2}\max\left(\frac{1}{g}|\sigma_{\mathrm{Px}}-\delta\theta_1|,\, g\,|\sigma_{\mathrm{Py}}-\delta\theta_2|\right)\Delta G_{\mathrm{p}} \\
& + B_{\mathrm{p}}\left[\frac{1}{2G_{\mathrm{p}}}\left|1-\frac{1}{g^2}\right|\phi_{\mathrm{a}}+\frac{1}{2G_{\mathrm{p}}}\left(\frac{1}{g^2}+1\right)\max\left(1,\,\phi_{\mathrm{a}}\right)+2\,\delta\phi_{12}\right]\Delta g \\
& + B_{\mathrm{a}}\,\frac{1}{G_{\mathrm{p}}}\max\left(\frac{1}{g^2}|\sigma_{\mathrm{Px}}-\delta\theta_1|,\, |\sigma_{\mathrm{Py}}-\delta\theta_2|\right)\Delta g \\
& + B_{\mathrm{a}}\,\frac{1}{G_{\mathrm{p}}}\max\left(\frac{\Delta\sigma_{\mathrm{Px}}}{g},\, g\,\Delta\sigma_{\mathrm{Py}}\right) \\
& + B_{\mathrm{a}}\,\frac{1}{G_{\mathrm{p}}}\max\left(\frac{\Delta(\delta\theta_1)}{g},\, g\,\Delta(\delta\theta_2)\right) \\
& + B_{\mathrm{p}}\,\frac{1}{2G_{\mathrm{p}}}\left[\left(\frac{1}{g}+g\right)+\left|g-\frac{1}{g}\right|\right]\Delta\phi_{\mathrm{a}} \\
& + B_{\mathrm{p}}\,2g\,\Delta(\delta\phi_{12}).
\end{aligned}
\tag{29}
$$

Again, as done in the calculation of the $X'$ component, we take the leading terms (the first order terms) and obtain a simplified expression of the error of residual component as:

$$
\begin{aligned}
|\Delta B_{Y'}| \leq\ & \Delta O_{\mathrm{S1/2}} \\
& + B_{\mathrm{p}}\left(\Delta G_{\mathrm{p}}+\Delta g+2\,\Delta(\delta\phi_{12})+\Delta\phi_{\mathrm{a}}\right) \\
& + B_{\mathrm{a}}\left(\Delta\sigma_{\mathrm{Px/y}}+\Delta(\delta\theta_{\mathrm{S1/2}})\right).
\end{aligned}
\tag{30}
$$

The differences from $\Delta B_{X'}$ (Eq. 27) are $2\Delta(\delta\phi_{12})$ and $\Delta\phi_{\mathrm{a}}$ in the second term in Eq. (30). The appearance of $\Delta\phi_{\mathrm{a}}$ means that the uncertainty of the magnetometer boom extension angle (the spin-plane rotation angle) causes a finite residual component,



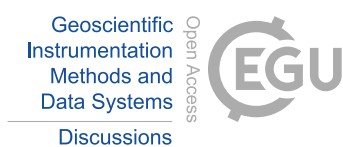

that is, the spin-plane ambient field is erroneously projected to yield the residual component $Y'$ by an angle of $\Delta\phi_a$. The effect of $\Delta\phi_a$ on the spin-plane primary field component is of the second order, while that on the residual component is of the first order. Accordig to to our estimate of the calibration parameter errors (Tab. 1), the first-order errors are in the range between $10^{-2}$ and $10^{-4}$ and the seconr-order errors (due to the couplings of calibration errors with the other small parameters) are in the range between $10^{-5}$ and $10^{-8}$.

## 2.3 Spin-axis component

The error of spin-axis component is derived from Eq. (19) in a straightforward fashion:

$$
\begin{aligned}
|\Delta B_{Z'}| \quad \leq \quad & \Delta O_3 + B_a\,\Delta\left(\frac{1}{G_a}\right) + B_p\,\max\left(\Delta\left(\frac{\sigma_{Px}}{G_a}\right),\,\Delta\left(\frac{\sigma_{Py}}{G_a}\right)\right) \qquad (31)\\
\leq \quad & \Delta O_3 \\
& + B_a\,\frac{1}{G_a^2}\Delta G_a \\
& + B_p\,\frac{1}{G_a^2}\max\left(\sigma_{Px},\sigma_{Py}\right)\,\Delta G_a \\
& + B_p\,\frac{1}{G_a}\max\left(\Delta\sigma_{Px},\,\Delta\sigma_{Py}\right) \qquad (32)
\end{aligned}
$$

For a nearly unit gain in the axial direction ($G_a \simeq 1$) and small misalignments ($\sigma_{Px} \ll 1$, $\sigma_{Py} \ll 1$), the expression of error estimate is simplified into:

$$
|\Delta B_{Z'}| \quad \leq \quad \Delta O_3 + B_a\,\Delta G_a + B_p\,\Delta\sigma_{Px/y}. \qquad (33)
$$

Equation (33) indicates that an error occurs in the spin-axis direction (1) when the offset $\Delta O_3$ is present, (2) when the axial (absolute) gain $G_a$ has an uncertainty, or (3) when the spin axis angle relative to the sensor $Z$ direction has an uncertainty (which introduces a mixing or projection of the spin-plane component by the spin-axis component).

## 3 Estimate of calibration parameter errors

Nominal errors (as upper limits) of calibration parameters are summarized in Tab. 1 as lessons from Earth-orbiting spinning spacecraft Cluster (Escoubet et al., 2001; Baloghet al., 2001) THEMIS (?Auster et al., 2008), and MMS (Burch et al., 2016; Russell et al., 2016). The spin-plane-related calibration parameters are assessed in detail by Plaschke et al. (2019). The accuracy studies on the spin-axis offset are presented by Alconcel et al. (2014), Frühauff et al. (2017), Plaschke (2019), and Schmid et al. (2020). In the following, we review the uncertainties of calibration parameters.

## 3.1 Offset error

The offsets in the spin plane ($O_1$ and $O_2$) are determined by the in-flight calibration. The error of spin-plane offsets on in-flight calibration is, after Plaschke et al. (2019), minimized down to the sum of (1) spin-plane component of natural fluctuation at





**Table 1.** Nominal errors of calibration parameters. Five lines from the top (spin axis angles, gain ratio, azimuthal angle, spin-plane offsets, and elevation angles) represent the in-flight calibration for THEMIS (Plaschke et al., 2019). Nominal error of spin-axis offset may vary between 0.2 nT in the solar wind (Plaschke, 2019) and 1 nT in the magnetosphere from temperature drift studies by Alconcel et al. (2014) and Frühauff et al. (2017). Absolute gains in the spin plane and along the spin axis are taken from the ground calibration experience. Spin-plane rotation angle is taken from the magnetometer boom design for BepiColombo Mio.

| Parameter | Symbol | Error upper limit |
|---|---|---|
| Spin axis angle (x or y directions) | $\Delta\sigma_{\mathrm{Px/y}}$ | $10^{-4}$ rad |
| Gain ratio | $\Delta g$ | $10^{-4}$ |
| Azimuthal angle | $\Delta(\delta\phi_{12})$ | $10^{-4}$ rad |
| Spin-plane offset S1 or S2 | $\Delta O_{\mathrm{S1/2}}$ | $0.1$ nT |
| Elevation angle S1 or S2 | $\Delta(\delta\theta_{\mathrm{S1/2}})$ | $10^{-3}$ rad |
| | | |
| Spin-axis offset S3 (solar wind) | $\Delta O_3^{(\mathrm{sw})}$ | $0.2$ nT |
| Spin-axis offset S3 (magnetosphere) | $\Delta O_3^{(\mathrm{ms})}$ | $1$ nT |
| Spin-plane absolute gain | $\Delta G_{\mathrm{p}}$ | $10^{-3}$ |
| Spin-axis absolute gain | $\Delta G_{\mathrm{a}}$ | $10^{-3}$ |
| Spin-plane rotation angle | $\Delta\phi_{\mathrm{a}}$ | $10^{-2}$ rad |

the spin frequency (denoted by $F_{\mathrm{p}}$), (2) projection of spin-axis ambient field by an error of spin-axis angle $B_{\mathrm{a}}\Delta\sigma_{\mathrm{Px/y}}$, and (3) projection of spin-axis ambient field by an error sensor elevation angle $B_{\mathrm{a}}\Delta(\delta\theta_{\mathrm{S1/2}})$:

$$\Delta O_{\mathrm{S1/2}} \simeq F_{\mathrm{p}} + B_{\mathrm{a}}\Delta\sigma_{\mathrm{Px/y}} + B_{\mathrm{a}}\Delta(\delta\theta_{\mathrm{S1/2}}). \tag{34}$$

The lesson from the in-flight calibration for the THEMIS magnetometer data indicates that an offset value of about 0.1 nT or better (i.e., smaller) can be reached using spacecraft spin (Plaschke et al., 2019).

The offset in the spin-axis direction cannot be determined from the spacecraft spin, but needs to be determined in different ways, for example, using additional measurements such as absolute magnetic field magnitude (Nakamura et al., 2014; Plaschke

et al., 2014) or using plasma physical properties such as the nearly-incompressible fluctuation nature in the solar wind (Hedgecock, 1975; Leinweber et al., 2008), the highly-compressible fluctuation nature in which the fluctuations are nearly aligned with the ambient field (Plaschke and Narita, 2016; Plaschke et al., 2017), or the magnetic null environment in diamagnetic cavities around comets (Goetz et al., 2016a, b). The uncertainty in the spin-axis offset can empirically be minimized to 0.2 nT when using the solar wind fluctuations (Plaschke, 2019) and the mirror-mode fluctuations (Plaschke and Narita, 2016; Frühauff

et al., 2017). The accuracy of spin-axis offset determination can be improved when a larger amount of data is available. An accuracy of 0.5 nT or 1.0 nT is considered as representative using the mirror-mode fluctuations (Schmid et al., 2020). It is also worth noting that the offset drift is up to 1 nT per year as lessons from Cluster (Alconcel et al., 2014) and THEMIS (Frühauff et al., 2017), which may be used as a nominal value of spin-axis offset error when the spacecraft stays in the magnetosphere and the in-situ offset determination using solar wind or mirror-mode fluctuations is not possible.





## 3.2 Gain error

The error of gain ratio in the spin plane is minimized to the natural fluctuation amplitude at the second harmonic of spin frequency in the spin plane (denoted by $F_{2\mathrm{p}}$) relative to the spin-plane ambient field $B_{\mathrm{p}}$ (Plaschke et al., 2019):

$$\Delta g \simeq \frac{F_{2\mathrm{p}}}{B_{\mathrm{p}}} \tag{35}$$

The gain ratio can be determined to a reasonably accurate level using the spacecraft spin, down to an uncertainty of about $10^{-4}$ (Plaschke et al., 2019). It is true that the gain ratio in the spin plane $g$ is related to the sensitivity measurements during the ground calibration through:

$$g^2 = \frac{S_{\mathrm{x}}}{S_{\mathrm{y}}}, \tag{36}$$

where $S_{\mathrm{x}}$ and $S_{\mathrm{y}}$ are the sensitivity (absolute gain) of the two spin-plane sensors, but the gain ratio obtained from the in-flight calibration is sufficiently accurate ($\Delta g \simeq 10^{-4}$) in practical applications.

## 3.3 Sensor axis non-orthogonality

Sensor axis non-orthogonality includes errors of the elevation angles $\Delta(\delta\theta_1)$ and $\Delta(\delta\theta_2)$ and azimuthal angles between S1 and S2 $\Delta(\delta\phi_{12})$. The error of elevation angles $\Delta(\delta\theta_1)$ and $\Delta(\delta\theta_2)$ is, after Plaschke et al. (2019), minimized to the sum of (1) natural frequency at the spin frequency relative to the ambient spin-axial field, (2) offset error relative to the ambient spin-axial field, and (3) uncertainty of the spin-axis angle as

$$\Delta(\delta\theta_{\mathrm{S1/2}}) \simeq \frac{F_{\mathrm{p}}}{B_{\mathrm{a}}} + \frac{\Delta O_{\mathrm{S1/2}}}{B_{\mathrm{a}}} + \Delta\sigma_{\mathrm{Px/y}}. \tag{37}$$

The elevation angles $\Delta(\delta\theta_1)$ and $\Delta(\delta\theta_2)$ are the angles between the sensors S1 and S3, and that between S2 and S3, respectively. The angle uncertainties $\Delta(\delta\theta_1)$ and $\Delta(\delta\theta_2)$ can be obtained both from the ground calibration and from the in-flight calibration. Errors of the elevation angles are about $10^{-3}$ in the in-flight calibration (Plaschke et al., 2019).

The azimuthal angle deviation $\delta\phi_{12}$ is also related to the ground-calibrated sensor angles $\xi_{12}$, $\xi_{13}$, and $\xi_{23}$. It is straightforward to show, by using the trigonometric relations, that the relation is

$$\sin(\delta\phi_{12}) = \sin(\delta\xi_{12}) + \sin(\delta\xi_{13})\sin(\delta\xi_{23}) \tag{38}$$

For smaller deviation angles of $\phi_{\mathrm{S12}}$, $\xi_{12}$, $\xi_{13}$, and $\xi_{23}$ (i.e., if the sensors are nearly orthogonal to one another), the relation is simplified into

$$\Delta(\delta\phi_{\mathrm{S12}}) \simeq \Delta(\delta\xi_{12}). \tag{39}$$

The azimuthal angle $\delta\phi_{\mathrm{S12}}$ can thus be obtained both from the ground calibration and from the in-flight calibration, and its uncertainty can be sufficiently minimized down to about $10^{-4}$ rad in the in-flight calibration (Plaschke et al., 2019).





### 3.4 Misalignment to the spacecraft reference direction

Angular deviation of the the spin axis from the normal direction of the sensor x–y plane is characterized by two angles, $\sigma_{\mathrm{Px}}$ and $\sigma_{\mathrm{Py}}$. The error of misalignment angles $\sigma_{\mathrm{Px}}$ and $\sigma_{\mathrm{Py}}$ is estimated as the ratio of the spin-axis natural fluctuation amplitude at the spin frequency to the spin-plane ambient field,

$$\sigma_{\mathrm{Px/y}} \simeq \frac{F_{\mathrm{a}}}{B_{\mathrm{p}}}, \tag{40}$$

and the value of $\sigma_{\mathrm{Px/y}}$ is empirically about $10^{-4}$ rad (Plaschke et al., 2019). The angles $\sigma_{\mathrm{Px}}$ and $\sigma_{\mathrm{Py}}$ need the determination or knowledge of spacecraft spin axis, and cannot usually be evaluated during the ground calibration of the sensors.

The remaining angle is the rotation angle in the spin plane The rotation angle can be determined in flight using Earth's magnetic field model in the case of Earth-orbiting spacecraft, and the method works better in a high-field environment. For example, the rotation angle is determined to an accuracy fo $0.5°$ or better when using the magnetic field data around the perigee with a field magnitude of about 8000 nT. In-flight determination of the rotation angle is meaningful when the accuracy in the in-flight method is better than the knowledge from the boom design with ground verification. We take the case of BepiColombo Mio magnetometer because the magnetometer boom extension direction is known to be within an uncertainty of $0.5°$ (which gives $\Delta\phi_{\mathrm{a}} = 8.7 \times 10^{-3}$ rad $\simeq 10^{-2}$ rad) from the spacecraft design and ground verification. As we will see in the next section, the uncertainty of rotation angle in the spin plane plays an important role in the final error estimate in a high-field environment.

### 4 Combined errors of calibrated magnetometer data

The individual error sources are combined using the first-order expressions (Eqs. 27, 30, and 33) to evaluate the error of calibrated magnetometer data for the nominal parameters (Tab. 1). Here, the errors represent the upper limits of the three magnetic field data in three directions (spin-plane primary, spin-plane residual, and spin-axis components). The combined errors are graphically displayed in Fig. 2 as a function of the ambient magnetic field in the spin-axis direction ($0°$, data curves in black) and spin-plane direction ($90°$, data curves in gray). Equations (27), (30), and (33) and Figure 2) indicate that the calibration error has two distinct domains: (1) the offset dominant domain in a low-field, up to an ambient field of about 1 nT when the field is along the spin axis (curves in black in Fig. 2), and up to 10 nT when the field is in the spin plane (curves in gray in Fig. 2), and (2) the ambient field-dependent domain in a high field (above 1 or 10 nT). In the low-field case, the offset dominates the magnetometer data error and the offset value is expected in the range between 0.1 to 1 nT. In the high-field case, the error grows linearly with the ambient field, and the relative error is expected between $1\%$ (which comes from $\Delta\phi_{\mathrm{a}}$) and $0.1\%$ (which comes from the absolute gain error and the elevation angle error).

The error depends on the angle between the ambient field and the spacecraft spin axis. The gain errors, azimuthal angle error, and boom misalignment are coupled to the spin-plane ambient field in the spin-plane components (Eqs. 27 and 30). The spin axis misalignment and elevation angle errors are coupled to the spin-axis field. The axial gain and the spin axis misalignment are coupled to the spin-axis and spin-plane ambient field, respectively, in the expression of spin-axis component (Eq. 33).




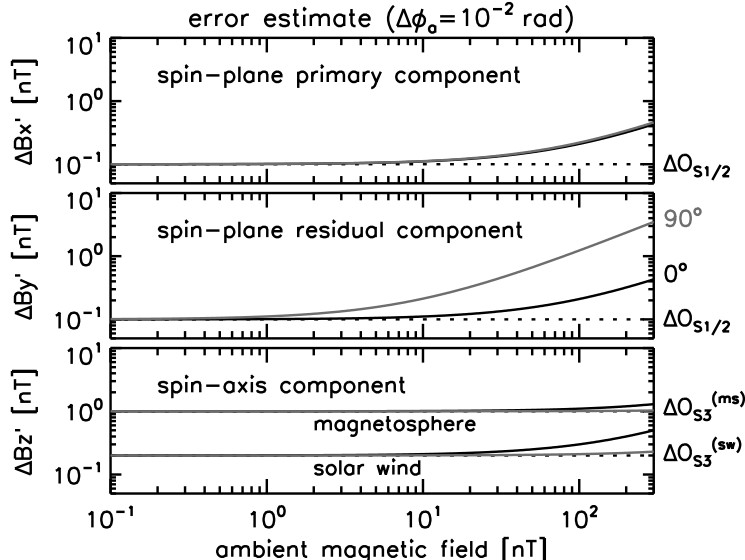

**Figure 2.** Error of in-flight calibrated magnetometer data for an error of magnetometer boom angle $\delta\phi_a \leq 0.5° \sim 10^{-2}$ rad (the case for the BepiColombo Mio magnetometer) Curves in black and in gray represent for the axial ambient magnetic field ($0°$ to the spin axis) and the spin-plane ambient field ($90°$), respectively.

The residual component has the largest uncertainty in Fig. 2, which comes from the uncertainty of spin-plane rotation angle $\Delta\phi_a$. For the reference purpose, Figure 3 exhibits the combined error estimate for the error of azimuthal angle smaller than that for Fig. 2 by an order of magnitude, $\delta\phi_a \sim 10^{-3}$ rad. In that case, the angle errors in the calibration parameters fall onto the nearly same order (between $10^{-4}$ rad and $10^{-3}$ rad). The final error is then below 1 nT (up to an ambient field of 300 nT) even when the ambient field is along the spin axis.

## 5 Conclusions

Fluxgate magnetometers are widely used in a wide range of spacecraft missions for the studies of Earth's and planetary magnetospheres, solar system bodies, and heliosphere. Magnetometer and the associated calibration process are necessarily accompanied by uncertainties that arise from various error sources. We conclude the error estimate on magnetometer in-flight calibration as follows.

1. Errors appear both as absolute ones (which are the offsets) and as relative ones (angle errors, gain errors). First-order expressions (Eqs. 27–33) (also graphically displayed in Figs. 2 and 3) are of practical use, and show that the offset errors dominate in a low ambient field (typically below 10 nT) while the relative errors (proportional to the ambient field) dominate in a high ambient field.

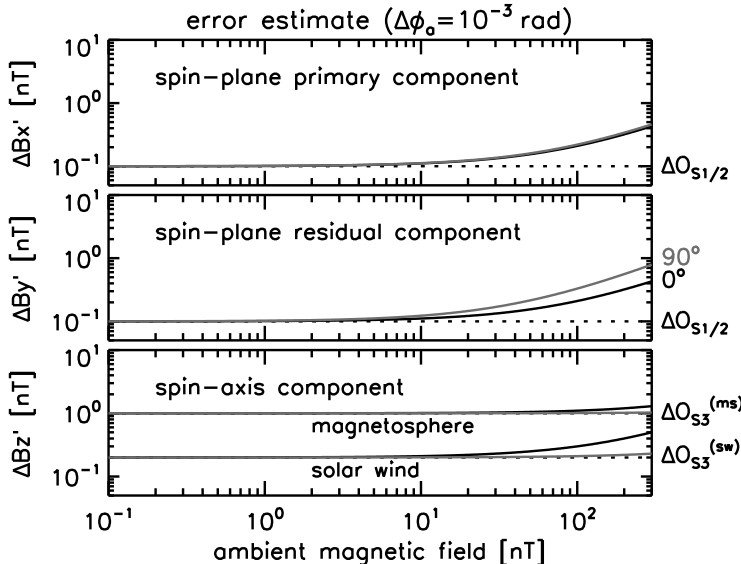

**Figure 3.** The same plot style as Fig. 2 but for the improved error of magnetometer boom angle $\delta\phi_a \leq 0.05° \sim 10^{-3}$ rad.

2. The largest uncertainty sources are (1) the spin-axis offset error and (2) the spin-plane rotation angle error. The offset error appears as the dominant error in the low-field environment The spin-plane rotation angle error plays a major role in a high-field environment, particularly when the ambient field is aligned with the spin axis.

The uncertainties are obtained by perturbing the calibration procedure proposed by Plaschke et al. (2019). When simplified into the first-order expression, the magnetometer data errors primarily represent the offset errors as constant and the errors of gains and angles as relative error to the ambient field. Our derivation sows how how the uncertainty sources combine through the calibration process both linearly (which is dominant) and non-linearly through coupling of calibration parameter errors (which is of only secondary importance when the errors of calibration parameters are small). The error formulas are

presented with analytical expressions (Eqs. 27, 30, and 33), and are expected to serve as a useful tool in various applications, for example, to further minimize the final error in designing a magnetometer with a boom and verifying the error throughly in the ground calibration (particularly the spin-plane rotation angle) and to report the error of scientific studies which are based on magnetometer data.

    It should be noted that the calibration parameters are treated as time independent in our study. In reality, however, the

315 calibration parameters (such as offsets and gains) depend on the temperature and can evolve along the orbit. Time-dependent picture of the calibration parameters needs an extensive in-flight calibration experience.

*Acknowledgements.* This work was financially supported by the Austrian Space Applications Programme (ASAP) at the Austrian Research Promotion Agency under contract 865967. YN also acknowledges financial support by the Japan Society for the Promotion of Science,





Invitational Fellowship for Research in Japan (long term) under grant FY2019 L19527. YN also thanks the research and administration staff
members in Hoshino laboratory group at the University of Tokyo for discussions, supports, and organizations during the fellowship program
and the University of Tokyo Mejirodai International Village (MIV) for the arrangement and hospitality during the pleasant and productive
stay in Tokyo.

## Appendix A: Derivatives

Detailed derivative calculations in section 2 are presented here.

$$325 \quad \Delta\left(\frac{1}{2G_{\mathrm{P}}}\left(\frac{1}{g}+g\right)\right) \;\leq\; \frac{1}{2G_{\mathrm{P}}^2}\left(\frac{1}{g}+g\right)\Delta G_{\mathrm{P}} + \frac{1}{2G_{\mathrm{P}}}\left|1-\frac{1}{g}\right|\Delta g \tag{A1}$$

$$\Delta\left(\frac{1}{gG_{\mathrm{P}}}\left(\sigma_{\mathrm{Px}}-\delta\theta_1\right)\right) \;=\; \Delta\left(\frac{1}{g}\right)\frac{1}{G_{\mathrm{P}}}\left(\sigma_{\mathrm{Px}}-\delta\theta_1\right)$$
$$+\frac{1}{g}\Delta\left(\frac{1}{G_{\mathrm{P}}}\right)\left(\sigma_{\mathrm{Py}}-\delta\theta_1\right)$$
$$+\frac{1}{gG_{\mathrm{P}}}\Delta\left(\sigma_{\mathrm{Px}}-\delta\theta_1\right) \tag{A2}$$
$$\leq\; \frac{1}{g^2 G_{\mathrm{P}}}\left|\sigma_{\mathrm{Px}}-\delta\theta_1\right|\Delta g$$
$$330 \qquad\qquad +\frac{1}{gG_{\mathrm{P}}^2}\left|\sigma_{\mathrm{Px}}-\delta\theta_1\right|\Delta G_{\mathrm{P}}$$
$$+\frac{1}{gG_{\mathrm{P}}}\Delta\sigma_{\mathrm{Px}} + \frac{1}{gG_{\mathrm{P}}}\Delta(\delta\theta_1) \tag{A3}$$

$$\Delta\left(\frac{g}{G_{\mathrm{P}}}\left(\sigma_{\mathrm{Py}}-\delta\theta_2\right)\right) \;=\; (\Delta g)\frac{1}{G_{\mathrm{P}}}\left(\sigma_{\mathrm{Py}}-\delta\theta_2\right)$$
$$+g\,\Delta\left(\frac{1}{G_{\mathrm{P}}}\right)\left(\sigma_{\mathrm{Py}}-\delta\theta_2\right)$$
$$+\frac{g}{G_{\mathrm{P}}}\Delta\left(\sigma_{\mathrm{Py}}-\delta\theta_2\right) \tag{A4}$$
$$335 \qquad\qquad \leq\; \frac{1}{G_{\mathrm{P}}}\left|\sigma_{\mathrm{Py}}-\delta\theta_2\right|\Delta g$$
$$+\frac{g}{G_{\mathrm{P}}^2}\left|\sigma_{\mathrm{Py}}-\delta\theta_2\right|\Delta G_{\mathrm{P}}$$
$$+\frac{g}{G_{\mathrm{P}}}\Delta\sigma_{\mathrm{Py}} + \frac{g}{G_{\mathrm{P}}}\Delta(\delta\theta_2<) \tag{A5}$$

$$\Delta\left(\frac{1}{2G_{\mathrm{P}}}\left(\frac{1}{g}-g\right)\right) \;=\; \Delta\left(\frac{1}{2G_{\mathrm{P}}}\right)\left(\frac{1}{g}-g\right) + \frac{1}{2G_{\mathrm{P}}}\Delta\left(\frac{1}{g}-g\right) \tag{A6}$$
$$\leq\; \frac{1}{2G_{\mathrm{P}}^2}\left|\frac{1}{g}-g\right|\Delta G_{\mathrm{P}} + \frac{1}{2G_{\mathrm{P}}}\left(\frac{1}{g^2}+1\right)\Delta g \tag{A7}$$





$$340 \quad \Delta\left(\frac{1}{2G_{\mathrm{p}}}\left(g-\frac{1}{g}\right)\phi_{\mathrm{a}}+g\,\delta\phi_{\mathrm{S}12}\right) \;=\; \Delta\left(\frac{1}{2G_{\mathrm{p}}}\right)\left(g-\frac{1}{g}\right)\phi_{\mathrm{a}}+\frac{1}{2G_{\mathrm{p}}}\Delta\left(g-\frac{1}{g}\right)\phi_{\mathrm{a}}$$

$$+\frac{1}{2G_{\mathrm{p}}}\left(g-\frac{1}{g}\right)\Delta\phi_{\mathrm{a}}+\delta\phi_{\mathrm{S}12}\,\Delta g+g\,\Delta(\delta\phi_{\mathrm{S}12}) \tag{A8}$$

$$\leq \quad \frac{1}{2G_{\mathrm{p}}^{2}}\left|g-\frac{1}{g}\right|\phi_{\mathrm{a}}\,\Delta G_{\mathrm{p}}+\frac{1}{2G_{\mathrm{p}}}\phi_{\mathrm{a}}\left(1+\frac{1}{g^{2}}\right)\Delta g$$

$$+\frac{1}{2G_{\mathrm{p}}}\left|g-\frac{1}{g}\right|\Delta\phi_{\mathrm{a}}+\delta\phi_{\mathrm{S}12}\,\Delta g+g\,\Delta(\delta\phi_{\mathrm{S}12}) \tag{A9}$$

$$\Delta\left(\frac{1}{2G_{\mathrm{p}}}\left(\frac{1}{g}+g\right)\phi_{\mathrm{a}}+g\,\delta\phi_{\mathrm{S}12}\right) \;=\; \Delta\left(\frac{1}{2G_{\mathrm{p}}}\right)\left(\frac{1}{g}+g\right)\phi_{\mathrm{a}}+\frac{1}{2G_{\mathrm{p}}}\phi_{\mathrm{a}}\Delta\left(\frac{1}{g}+g\right)$$

$$345 \qquad\qquad +\frac{1}{2G_{\mathrm{p}}}\left(\frac{1}{g}+g\right)\Delta\phi_{\mathrm{a}}+\delta\phi_{\mathrm{S}12}\,\Delta g+g\Delta(\delta\phi_{\mathrm{S}12}) \tag{A10}$$

$$\leq \quad \frac{1}{2G_{\mathrm{p}}^{2}}\left(\frac{1}{g}+g\right)\phi_{\mathrm{a}}\,\Delta G_{\mathrm{p}}+\frac{1}{2G_{\mathrm{p}}}\left(1-\frac{1}{g^{2}}\right)\Delta g$$

$$+\frac{1}{2G_{\mathrm{p}}}\left(\frac{1}{g}+g\right)\Delta\phi_{\mathrm{a}}+\delta\phi_{\mathrm{S}12}\,\Delta g+g\,\Delta(\delta\phi_{\mathrm{S}12}) \tag{A11}$$



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
