# Peer review of "Error estimate for fluxgate magnetometer in-flight calibration on a spinning spacecraft"

_Geoscientific Instrumentation, Methods and Data Systems, 2020_

## Referee Comment (RC1) · Anonymous Referee #1 · 2 Nov 2020

**1   General remarks**

The submitted manuscript provides an analysis of the errors of the FGM magnetic field data related to the uncertainties of the calibration parameters determined for spinning spacecraft through the procedure described by Plaschke et al. 2019. The errors are computed in the first order by introducing small perturbations of the calibration parameters and taking into account typical values of the uncertainties as determined for a number of spacecraft (Cluster, THEMIS, MMS, and BepiColombo).

The manuscript can be a useful resource both for planning future spacecraft missions and for estimating the accuracy of the magnetic field data delivered by current missions. There are however a number of issues which should be addressed before publication.

[Figure]

**2 Specific comments**

• page 2, line 29-30: Please explain what do you mean through "Fourier series of space-craft spin frequency".

• page 2, line 45-49: Is $\phi_a$ the angle between the projection of the spin axis on the $(P_x, P_y)$-plane of the sensor package coordinate system? If so, please mention this in the description of the step 3.

• page 2, line 48: "... $\sigma_{P_x}$ and $\sigma_{P_y}$ (with respect to the $P_z$ axis) ..." This definition of $\sigma_{P_x}$ and $\sigma_{P_y}$ is nor clear enough. From Fig. 1 it appears that $\sigma_{P_y}$ is the angle between the $P_z$ axis and the projection of the spin axis on the $(P_z, P_y)$-plane. $\sigma_{P_x}$ appears to be the angle between the spin axis and the $(P_z, P_y)$-plane. If this is correct, please clarify this in the description of step 3.

• page 5, line 55-59: The description of step 5 is misleading. A clear distinction should be made between the ideal exact transformations and the transformations derived through the calibration procedure. The "above transformations" as described in steps 1-4 are the exact transformations (not derived through calibration, thus not affected by calibration errors). If they are inverted then the true field is obtained. In lines 56-59 the Authors refer to the transformations estimated through the calibration procedure. To eliminate the potential confusion I suggest that at each step the corresponding transformation is explicitly stated. E.g. step 2: coord 1 $\mathbf{\Omega}^{-1}$ coord 2; step 3: coord 2 $\mathbf{\Sigma}^{-1}\mathbf{\Phi}^{-1}$ coord 3; step 4: coord 3 $\mathbf{G}^{-1}\mathbf{\Gamma}^{-1} + \mathbf{O}_s$ coord 4; step 5: coord 4 $\mathbf{\Omega}'\mathbf{\Phi}'\mathbf{\Sigma}'\mathbf{\Gamma}'\mathbf{G}' - \mathbf{O}'_s$ coord 5; direct transform: coord 4 $\mathbf{\Omega}\mathbf{\Phi}\mathbf{\Sigma}\mathbf{\Gamma}\mathbf{G} - \mathbf{O}_s$ coord 1; The prime symbol denotes the quantities derived using the calibration procedure. Alternatively, the transformations could be added to Fig. 1.

• page 3, line 55,60-61: While correct, the formulations in lines 55 "... transformations are inverted to estimate the magnetic field ..." taken together with the lines 60-61, "... the forward transform is defined for the conversion ..." can be confusing for the reader. Adding the transforms to the steps as suggested above and reformulating line 55 could help clarifying this.

• page 4, line 82: "(angle between the coord 2 system and the coord 3 system)" formulation is ambiguous. Please reformulate.

• page 4, line 84-86: The descriptions of $\delta\theta_1$, $\delta\theta_2$ and $\delta\Phi_{12}$ are unclear. Perhaps instead of "with a relation to the deviation from $90°$", simply writing "the deviation from $90°$ of the elevation/azimuth" angles would be better.

• page 5, line 115-118: The formulation is difficult to understand. Please reformulate to make clear that the coord 5 system is obtained using the transformation matrices resulted from the calibration procedure.

• page 12, section 4: It might be useful to rewrite the estimated errors (Eqs. 27,30,33) using the expected values given in Table 1.

$|\Delta B_{x'}| \leq 0.1nT + (B_p + B_a) \times 10^{-2}$
$|\Delta B_{y'}| \leq 0.1nT + (10B_p + B_a) \times 10^{-3}$
$|\Delta B_{z'}| \leq 0.21nT + (B_p + B_a) \times 10^{-3}$

or similar.

• page 13,14, Fig 2,3: Many spacecraft measure magnetic fields much higher than 300 nT. Please extend the ambient magnetic field domain ($x$-axis) to 10 000 nT.

• page 14, lines 303-304: "... when the ambient field is aligned with the spin axis ...". From Eq.(30) $\Phi_a$ multiplies $B_p$, therefore the largest error due to $\Phi_a$ is for the ambient magnetic field orthogonal to the spin axis.

**3  Minor points**

• page 1, line 5 and elsewhere in the manuscript: "... perturbing the calibration procedure ..." – not the procedure, but the calibration parameters are perturbed.

• page 1, line 7-8: "The final error ... are important factors." this sentence is difficult to understand. Please reformulate.

• page 2, line 31: "sensor output", "not ourput"

• page 2, line 32: "sensor-axes" (if plural)

• page 4, line 72: "The", not "There"

• page 4, line 82: remove "to the angles"

• page 13,14, Fig 2,3: The gray lines cannot be distinguished from the black lines. Please use colors or solid/dashed lines.

• page 14, line 307: remove one "how"

---

## Referee Comment (RC2) · Anonymous Referee #2 · 19 Nov 2020

Full disclosure: this reviewer also reviewed Plaschke et al 2019.

While starting this review I had some concern there would be a large overlap with the earlier paper, and that my eyes would not be "fresh" enough. Happily this is not the case. The review of the earlier paper has the right level of detail, enough to provide a good picture without being excessive. The first reviewer provides a good discussion of this part of the present manuscript so I leave it at that.

I agree with the first reviewer, that the grey line/black line markings in Fig's 2 and 3 are difficult to observe.

Finally, it appears that this study of errors is targeted at low-field measurement settings. If this is the case is should be made clear. There is also no discussion of magnetometer

nonlinearity errors as distinct from the errors due to the nonlinear coupling of the parameters under consideration. The latter is suitably noted in the paper, and in low-field settings sensor non linearity is usually negligible.

In low earth orbit this can be different. Modern sensors which are often double wound, and even triple wound have excellent linearity, but not always. The 1979 MAGSAT single-wound sensor suffered from about 1% nonlinearity, and the same sensor design was used more recently on MESSENGER. With present thinking about the possibility of deploying large fleets of small magnetometer cubesats with just as small sensors one might ask whether nonlinearity issues can rise again. Also the potential quantity of spacecraft points to wanting automated calibration and error estimation methods. Perhaps the methods of Plaschke at al 2019 and the present paper will make that task easier.

Here is a small collection of additional minor corrections:

2-31 "sensor ourput"

9-189 "seconr order"

9-206 "?Auster"

12-266 "accuracy fo"

12-277 "Figure 2)"

14-307 "sows how how"

14-311 "error throughly"

---

## Author Comment (AC1) · 10 Dec 2020

**Reply to referee comments**

Manuscript ID: GI–2020–31
Error estimate for fluxgate magnetometer in-flight calibration on a spinning
spacecraft
Y. Narita, F. Plaschke, W. Magnes, D. Fischer, and D. Schmid
* * *
**Referee 1**

1. *1. General remarks*

   *The submitted manuscript provides an analysis of the errors of the FGM magnetic field data related to the uncertainties of the calibration parameters determined for spinning spacecraft through the procedure described by Plaschke et al. 2019. The errors are computed in the first order by introducing small perturbations of the calibration parameters and taking into account typical values of the uncertainties as determined for a number of spacecraft (Cluster, THEMIS, MMS, and BepiColombo).*

   *The manuscript can be a useful resource both for planning future spacecraft missions and for estimating the accuracy of the magnetic field data delivered by current missions. There are however a number of issues which should be addressed before publication.*

   **Reply**:

   - Thank you for careful reading, thoughtful comments, and positive evaluation.

2. *2. Specific comments*

   *page 2, line 29–30: Please explain what do you mean through "Fourier series of spacecraft spin frequency".*

   **Reply**:

   - We changed the text as follows to avoid confusion.

   - page 2, line 31–35
     "...Fourier series over the frequencies as

     $$B_i(t) \;=\; \sum_{n=0}^{N-1} F_i(\omega)\, \mathrm{e}^{in\omega t}$$

     for the $i$-th component of magnetic field. $F_i$ is the Fourier coefficient, i the imaginary unit, $\omega$ the de-spinning frequency (as angular frequency), $N$ the number of data points, and $t$ the time in the data."

3. *page 2, line 45–49: Is $\phi_a$ the angle between the projection of the spin axis on the $(Px, Py)$-plane of the sensor package coordinate system? If so, please mention this in the description of the step 3.*

   **Reply**:

   - No, $\phi_\mathrm{a}$ is the azimuthal angle around the spacecraft spin axis in the xy-plane in the coord-2 system.

   - page 3, line 56–59
     "first by rotating ... (with the rotation angle $\phi_\mathrm{a}$ in the xy-plane around the spin axis in the coord-2 system) and then by orienting..."

4. *page 2, line 48: "... $\sigma_{Px}$ and $\sigma_{Py}$ (with respect to the $Pz$ axis) ..." This definition of $\sigma_{Px}$ and $\sigma_{Py}$ is nor clear enough. From Fig. 1 it appears that $\sigma_{Py}$ is the angle between the $Pz$ axis and the projection of the spin axis on the $(Pz, Py)$-plane. $\sigma_{Px}$ appears to be the angle between the spin axis and the $(Pz, Py)$-plane. If this is correct, please clarify this in the description of step 3.*

   **Reply**:

   - The definition of the angles $\sigma_{Px}$ and $\sigma_{Py}$ suggested by the referee is perfect and unambiguous. Thank you for the suggestion. We included the definition.

   - page 3, line 61–62
     "Here, $\sigma_{Py}$ is the angle between the $Pz$ axis and the projection of the spin axis on the $(Pz, Py)$-plane. $\sigma_{Px}$ is the angle between the spin axis and the $(Pz, Py)$-plane."

5. *page 5, line 55–59: The description of step 5 is misleading. A clear distinction should be made between the ideal exact transformations and the transformations derived through the calibration procedure. The "above transformations" as described in steps 1–4 are the exact transformations (not derived through calibration, thus not affected by calibration errors). If they are inverted then the true field is obtained. In lines 56–59 the Authors refer to the transformations estimated through the calibration procedure. To eliminate the potential confusion I suggest that at each step the corresponding transformation is explicitly stated. E.g. step 2: coord 1 $\Omega^{-1}$ coord 2; step 3: coord 2 $\Sigma^{-1}\Phi^{-1}$ coord 3; step 4: coord 3 $G^{-1}\Gamma^{-1} + O_s$ coord 4; step 5: coord 4 $\Omega'\Phi'\Sigma'\Gamma'G' - O_s^p rime$ coord 5; direct transform: coord 4 $\Omega\Phi\Sigma\Gamma G - O_s$ coord 1; The prime symbol denotes the quantities derived using the calibration procedure. Alternatively, the transformations could be added to Fig. 1.*

**Reply**:

- Good idea. We added equations in each step to make the explanations unambiguous.

- page 2, line 43–44
  "... The magnetic field modeled in the coord-1 system as

$$\boldsymbol{B}_{c1} = \begin{bmatrix} B_X \\ B_Y \\ B_Z \end{bmatrix} = \begin{bmatrix} B_p \\ 0 \\ B_a \end{bmatrix}$$

  ..."

- page 3, line 53–55
  "...The magnetic field vector in the coord-2 system is symbolically related to that in the coord-1 system as

$$\boldsymbol{B}_{c2} = \boldsymbol{\Omega}^{-1}\, \boldsymbol{B}_{c1},$$

  where $\boldsymbol{\Omega}^{-1}$ is the spin rotation matrix Note that $\boldsymbol{\Omega}$ is defined as the despinning matrix here."

- page 3, line 62–67
  "...The magnetic field vector in the coord-3 system is symbolically related to that in the coord-2 system as

$$\boldsymbol{B}_{c3} = \Sigma^{-1}\, \boldsymbol{\Phi}^{-1}\, \boldsymbol{B}_{c2},$$

  where $\boldsymbol{\Phi}^{-1}$ is the azimuthal rotation matrix in the spin plane (around the spin axis in the coord-2 system) and $\boldsymbol{\Sigma}^{-1}$ is the transformation matrix to orient the z axis in the direction to the sensor package Pz direction. Again the matrices without inversion are used for the reconstruction of the model magnetic field in the calibration."

- page 3, line 72–76
  "...The magnetic field vector in the coord-4 system is symbolically related to that in the coord-3 system as

$$\boldsymbol{B}_{c4} = \mathbf{G}^{-1}\boldsymbol{\Gamma}^{-1}\, \boldsymbol{B}_{c3} + \boldsymbol{O}_s,$$

  where $\boldsymbol{\Gamma}^{-1}$ is the transformation matrix using three angles ($\theta_1$, $\theta_2$, and $\phi_{12}$), $\mathbf{G}^{-1}$ is the gain matrix, and $\boldsymbol{O}_s$ is the offset vector."

6. *page 3, line 55, 60–61: While correct, the formulations in lines 55 "... trans-formations are inverted to estimate the magnetic field ..." taken together with the lines 60-61, "... the forward transform is defined for the conversion ..." can be confusing for the reader. Adding the transforms to the steps as suggested above and reformulating line 55 could help clarifying this.*

   **Reply**:

   - We added an equation for the reconstruction of the model field using the sensor magnetic field and the calibration matrices.

   - page 4, line 81–84
     "... The model magnetic field is reconstructed from the sensor magnetic field as

     $$\boldsymbol{B}_{c5} = \boldsymbol{\Omega}\,\boldsymbol{\Phi}\,\boldsymbol{\Sigma}\,\boldsymbol{\Gamma}\,\mathbf{G}\,(\boldsymbol{B}_{c4} - \boldsymbol{O}_s).$$

     If the calibration parameters are all known, the reconstructed field $\boldsymbol{B}_{c5}$ restores the model field $\boldsymbol{B}_{c1}$."

   - page 4, line 89–92
     "...The relation between the sensor-output magnetic field $\boldsymbol{B}_s = \boldsymbol{B}_{c4}$ (introduced in the coord-4 system) and the model ambient field in the spinning frame $\boldsymbol{B}_{c2}$ (introduced in the coord-2 system, Eqs. 3–5) is expressed by a set of transformation matrices $\mathbf{G}^{-1}\,\boldsymbol{\Gamma}^{-1}\,\boldsymbol{\Sigma}^{-1}\,\boldsymbol{\Phi}^{-1}$ and an offset vector $\boldsymbol{O}_s$ as (Plaschke et al., 2019)

     $$\boldsymbol{B}_s = \mathbf{G}^{-1}\,\boldsymbol{\Gamma}^{-1}\,\boldsymbol{\Sigma}^{-1}\,\boldsymbol{\Phi}^{-1}\boldsymbol{B}_{c2} + \boldsymbol{O}_s.$$

     ..."

7. *page 4, line 82: "(angle between the coord 2 system and the coord 3 system)" formulation is ambiguous. Please reformulate.*

   **Reply**:

   - We changed the text as follows.

   - page 5, line 108–110
     "...in sensor package system ($\sigma_{Py}$ is the angle between the sensor-3 direction and the projection of the spin axis onto the sensor package Py–Pz plane; $\sigma_{Px}$ is the angle of spin axis and the sensor package Py–Pz plane)"

8. *page 4, line 84–86: The descriptions of $\delta\theta_1$, $\delta\theta_2$ and $\delta\Phi_{12}$ are unclear. Perhaps instead of "with a relation to the deviation from 90°", simply writing "the deviation from 90° of the elevation/azimuth" angles would be better.*

   **Reply**:

- Rewritten as follows.

- page 5, line 111
  "deviation of elevation angles from $90°$ defined as $\delta\theta_1$ and $\delta\theta_2$, for the sensors 1 and 2, respectively"

- page 5, line 112
  "deviation of azimuthal angle from $90°$ defined as $\delta\phi_{12}$"

9. *page 5, line 115–118: The formulation is difficult to understand. Please reformulate to make clear that the coord 5 system is obtained using the transformation matrices resulted from the calibration procedure.*

   **Reply**:

   - We reformulated as follows.

   - page 6, line 141–143
     "Here, the magnetic field vector $(B_{X'}, B_{Y'}, B_{Z'})$ is represented in the coord-5 system and hence ideally reproduce the model magnetic field in the coord-1 system. That is, the z-component is in the direction of spacecraft spin axis, the x-component is is in the spin plane. The y-component is also in the spin plane but should ideally not contain the ambient field."

10. *page 12, section 4: It might be useful to rewrite the estimated errors (Eqs. 27, 30, 33) using the expected values given in Table 1.*

$$\begin{aligned}
|\Delta B_{x'}| &\leq 0.1nT + (B_p + B_a) \times 10^{-2} \\
|\Delta B_{y'}| &\leq 0.1nT + (10B_p + B_a) \times 10^{-3} \\
|\Delta B_{z'}| &\leq 0.21nT + (B_p + B_a) \times 10^{-3}
\end{aligned}$$

*or similar.*

   **Reply**:

   - Good idea!

   - page 13, line 300–305
     "... For a practical purpose, the combined errors in Eqs. (33), (36), and (39) are reformulated in an approximate form using the values given in

Tab. 1:

$$
\begin{aligned}
|\Delta B_{x'}| &\le 0.1 \ [\text{nT}] + (B_\mathrm{p} + B_\mathrm{a}) \times 10^{-2} \\
|\Delta B_{y'}| &\le 0.1 \ [\text{nT}] + (10B_\mathrm{p} + B_\mathrm{a}) \times 10^{-3} \\
|\Delta B_{z'}| &\le 0.2 \ [\text{nT}] + (B_\mathrm{p} + B_\mathrm{a}) \times 10^{-3}.
\end{aligned}
$$

..."

11. *page 13, 14, Fig 2, 3: Many spacecraft measure magnetic fields much higher than 300 nT. Please extend the ambient magnetic field domain (x-axis) to 10 000 nT.*

    **Reply**:

    - The plots were extended to ambient field to 10,000 nT and were added as Fig. 4 and Fig. 5. We still keep the original low-field case (Fig. 2 and Fig. 3, up to 300 nT) for the potential use to the BepiColombo Mio spacecraft.
    - Figures 4 and 5 (pages 15 and 16).

12. *page 14, lines 303–304: "... when the ambient field is aligned with the spin axis ...". From Eq.(30) $\Phi_a$ multiplies $B_p$ , therefore the largest error due to $\Phi_a$ is for the ambient magnetic field orthogonal to the spin axis.*

    **Reply**:

    - True! Thank you!

    - page 16, line 336
      "...ambient field is aligned in the spin plane."

13. *3. Minor points*

    *page 1, line 5 and elsewhere in the manuscript: "... perturbing the calibration procedure ..". — not the procedure, but the calibration parameters are perturbed.*

    **Reply**:

    - Right! Changed into "calibration parameters." (page 1, line 5; page 16, line 337).

14. *page 1, line 7–8: "The final error ... are important factors." this sentence is difficult to understand. Please reformulate.*

    **Reply**:

- The sentence was simplified.

- page 1, line 7–8
  "The final error depends on (1) the magnitude of magnetic field with respect to the offset error and (2) the angle of magnetic field to the spacecraft spin axis."

15. *page 2, line 31: "sensor output", "not ourput"*

    **Reply**:

    - Done (page 2, line 36).

16. *page 2, line 32: "sensor-axes" (if plural)*

    **Reply**:

    - Done. (page 2, line 37; page 5, line 98)

17. *page 4, line 72: "The", not "There"*

    **Reply**:

    - Done. "The matrices" on page 5, line 98. It was a misspelling of "These" but I find "The" also appropriate.

18. *page 4, line 82: remove "to the angles"*

    **Reply**:

    - Done (page 5, line 108).

19. *page 13, 14, Fig 2, 3: The gray lines cannot be distinguished from the black lines. Please use colors or solid/dashed lines.*

    **Reply**:

    - The gray curve is re-plotted with dashed line in darker, thicker gray in the revision (Fig. 2 and Fig. 3, pages 14 and 15).

20. *page 14, line 307: remove one "how"*

    **Reply**:

- Done. "shows how" (page 16, line 339).
* * *
**Referee 2**

1. *Full disclosure: this reviewer also reviewed Plaschke et al 2019.*

   *While starting this review I had some concern there would be a large overlap with the earlier paper, and that my eyes would not be "fresh" enough. Happily this is not the case. The review of the earlier paper has the right level of detail, enough to provide a good picture without being excessive. The first reviewer provides a good discussion of this part of the present manuscript so I leave it at that.*

   *I agree with the first reviewer, that the grey line/black line markings in Figs 2 and 3 are difficult to observe.*

   *Finally, it appears that this study of errors is targeted at low-field measurement settings. If this is the case is should be made clear. There is also no discussion of magnetometer nonlinearity errors as distinct from the errors due to the nonlinear coupling of the parameters under consideration. The latter is suitably noted in the paper, and in low-field settings sensor non linearity is usually negligible.*

   *In low earth orbit this can be different. Modern sensors which are often double wound, and even triple wound have excellent linearity, but not always. The 1979 MAGSAT single-wound sensor suffered from about 1% nonlinearity, and the same sensor design was used more recently on MESSENGER. With present thinking about the possibility of deploying large fleets of small magnetometer cubesats with just as small sensors one might ask whether nonlinearity issues can rise again. Also the potential quantity of spacecraft points to wanting automated calibration and error estimation methods. Perhaps the methods of Plaschke at al 2019 and the present paper will make that task easier.*

   **Reply**:

   - Figs. 2 and 3 were improved using darker, thicker, dashed lines (page 14, page 15).

   - We address the scope of manuscript more clearly in the introduction.

   - page 1, line 23 to page 2, line 26
     "The scope of our work is the error estimate of calibrated magnetometer data in a low-field environment. In practice, more effects need to be taken into account, including sensor nonlinearities, temperature dependence (temperature drift effect), and jumps in the data associated with the change in operational modes."

- We agree with the importance of nonlineariry issues. We added the following sentences at the end of manuscript.

- page 16, line 349 to page 17, line 356
  "The errors associated with the uncertainties in calibration parameters are studied in this paper. In a low-field environment such as in interplanetary space the sensor nonlinearity (which originates in the nonlinearity of gain) is usually considered negligible. In a low Earth orbit the situation may be different. Modern sensors which are often double wound, and even triple wound have excellent linearity (typically to an accuracy of about $10^{-4}$ per axis), but this is not always the case. The MAGSAT single-wound sensor (Acuña, 1980; Langel et al., 1982), for example, suffered from about 1% nonlinearity, and the same sensor design was used more recently on MESSENGER (Solomon et al., 2007; Anderson et al., 2007). With present thinking about the possibility of deploying large fleets of small magnetometer cubesats with just as small sensors one might ask whether nonlinearity issues can rise again."

2. *Here is a small collection of additional minor corrections:*
   *2–31 "sensor ourput"*

   **Reply**:

   - Corrected to "sensor output" (page 2, line 36).

3. *9–189 "seconr order"*

   **Reply**:

   - Corrected to "second order" (page 9, line 212).

4. *9–206 "?Auster"*

   **Reply**:

   - Corrected to "THEMIS (Angelopoulos, 2008; Auster et al., 2008)" (page 10, line 231).

5. *12–266 "accuracy fo"*

   **Reply**:

   - Corrected to "accuracy of" (page 13, line 291).

6. *12–277 "Figure 2)"*

   **Reply**:

   - Corrected to "Fig. 2" (page 13, line 307).

7. *14–307 "sows how how"*

   **Reply**:

   - Corrected to "show how" (page 16, line 339).

8. *14–311 "error throughly"*

   **Reply**:

   - Corrected to "error thoroughly" (page 16, line 343).
* * *
**Other changes**

The following literatures were added:

[revised manuscript text omitted]
{\sigma_{\mathrm{Px}}}{G_\mathrm{a}}\right), \, \Delta\left(\frac{\sigma_{\mathrm{Py}}}{G_\mathrm{a}}\right)\right) \quad (37) \\
&\leq \Delta O_3 \\
&\quad + B_\mathrm{a} \frac{1}{G_\mathrm{a}^2}\Delta G_\mathrm{a} \\
&\quad + B_\mathrm{p} \frac{1}{G_\mathrm{a}^2}\max\left(\sigma_{\mathrm{Px}}, \sigma_{\mathrm{Py}}\right)\Delta G_\mathrm{a} \\
&\quad + B_\mathrm{p} \frac{1}{G_\mathrm{a}}\max\left(\Delta\sigma_{\mathrm{Px}}, \Delta\sigma_{\mathrm{Py}}\right) \quad (38)
\end{aligned}
$$

For a nearly unit gain in the axial direction ($G_\mathrm{a} \simeq 1$) and small misalignments ($\sigma_{\mathrm{Px}} \ll 1$, $\sigma_{\mathrm{Py}} \ll 1$), the expression of error estimate is simplified into:

$$
|\Delta B_{Z'}| \leq \Delta O_3 + B_\mathrm{a} \, \Delta G_\mathrm{a} + B_\mathrm{p} \, \Delta\sigma_{\mathrm{Px/y}}. \quad (39)
$$

[revised manuscript text omitted]